# Information-theoretic Limits for Community Detection in Network Models

**Chuyang Ke**
Department of Computer Science
Purdue University
West Lafayette, IN 47907
cke@purdue.edu

**Jean Honorio**
Department of Computer Science
Purdue University
West Lafayette, IN 47907
jhonorio@purdue.edu

## Abstract

We analyze the information-theoretic limits for the recovery of node labels in several network models. This includes the Stochastic Block Model, the Exponential Random Graph Model, the Latent Space Model, the Directed Preferential Attachment Model, and the Directed Small-world Model. For the Stochastic Block Model, the non-recoverability condition depends on the probabilities of having edges inside a community, and between different communities. For the Latent Space Model, the non-recoverability condition depends on the dimension of the latent space, and how far and spread are the communities in the latent space. For the Directed Preferential Attachment Model and the Directed Small-world Model, the non-recoverability condition depends on the ratio between homophily and neighborhood size. We also consider dynamic versions of the Stochastic Block Model and the Latent Space Model.

## 1 Introduction

Network models have already become a powerful tool for researchers in various fields. With the rapid expansion of online social media including *Twitter*, *Facebook*, *LinkedIn* and *Instagram*, researchers now have access to more real-life network data and network models are great tools to analyze the vast amount of interactions [16, 2, 1, 21]. Recent years have seen the applications of network models in machine learning [5, 33, 23], bioinformatics [9, 15, 11], as well as in social and behavioral researches [26, 14].

Among these literatures one of the central problems related to network models is community detection. In a typical network model, nodes represent individuals in a social network, and edges represent interpersonal interactions. The goal of community detection is to recover the label associated with each node (i.e., the community where each node belongs to). The exact recovery of 100% of the labels has always been an important research topic in machine learning, for instance, see [2, 10, 20, 27].

One particular issue researchers care about in the recovery of network models is the relation between the number of nodes, and the proximity between the likelihood of connecting within the same community and across different communities. For instance, consider the Stochastic Block Model (SBM), in which $p$ is the probability for connecting two nodes in the same community, and $q$ is the probability for connecting two nodes in different communities. Clearly if $p$ equals $q$, it is impossible to identify the communities, or equivalently, to recover the labels for all nodes. Intuitively, as the difference between $p$ and $q$ increases, labels are easier to be recovered.

In this paper, we analyze the information-theoretic limits for community detection. Our main contribution is the comprehensive study of several network models used in the literature. To accomplish that task, we carefully construct restricted ensembles. The key idea of using restricted ensembles is that for any learning problem, if a subclass of models is difficult to be learnt, then the original class

Table 1: Comparison of network models (S - static; UD - undirected dynamic; DD - directed dynamic)

| Type | Model | Our Result | Previous Result | Thm. No. |
|------|-------|------------|-----------------|----------|
| S | SBM | $\frac{(p-q)^2}{q(1-q)} \leq \frac{2\log 2}{n} - \frac{4\log 2}{n^2}$ | $\frac{(p-q)^2}{p+q} \leq \frac{2}{n}$[25] <br> $\frac{(p-q)^2}{q(1-q)} \leq O(\frac{1}{n})$[10] | Thm. 1 |
| S | ERGM | $2(\cosh\beta - 1) \leq \frac{2\log 2}{n} - \frac{4\log 2}{n^2}$ | Novel | Cor. 1 |
| S | LSM | $(4\sigma^2 + 1)^{-1-p/2}\|\mu\|_2^2 \leq \frac{\log 2}{2n} - \frac{\log 2}{n^2}$ | Novel | Thm. 2 |
| UD | DSBM | $\frac{(p-q)^2}{q(1-q)} \leq \frac{(n-2)\log 2}{n^2-n}$ | Novel | Thm. 3 |
| UD | DLSM | $(4\sigma^2 + 1)^{-1-p/2}\|\mu\|_2^2 \leq \frac{(n-2)\log 2}{4n^2-4n}$ | Novel | Thm. 4 |
| DD | DPAM | $(s+1)/8m \leq 2^{(n-2)/(n^2-n)}/n^2$ | Novel | Thm. 5 |
| DD | DSWM | $(s+1)^2/(mp(1-p)) \leq 2^{2(n-2)/n^2}/n$ | Novel | Thm. 6 |

of models will be at least as difficult to be learnt. The use of restricted ensembles is customary for information-theoretic lower bounds [28, 31].

We provide a series of novel results in this paper. While the information-theoretic limits of the Stochastic Block Model have been heavily studied (in slightly different ways), none of the other models considered in this paper have been studied before. Thus, we provide new information-theoretic results for the Exponential Random Graph Model (ERGM), the Latent Space Model (LSM), the Directed Preferential Attachment Model (DPAM), and the Directed Small-world Model (DSWM). We also provide new results for dynamic versions of the Stochastic Block Model (DSBM) and the Latent Space Model (DLSM).

Table 1 summarizes our results.

## 2 Static Network Models

In this section we analyze the information-theoretic limits for two static network models: the Stochastic Block Model (SBM) and the Latent Space Model (LSM). Furthermore, we include a particular case of the Exponential Random Graph Model (ERGM) as a corollary of our results for the SBM. We call these static models, because in these models edges are independent of each other.

### 2.1 Stochastic Block Model

Among different network models the Stochastic Block Model (SBM) has received particular attention. Variations of the Stochastic Block Model include, for example, symmetric SBMs [3], binary SBMs [27, 13], labelled SBMs [36, 20, 34, 18], and overlapping SBMs [4]. For regular SBMs [25] and [10] showed that under certain conditions recovering the communities in a SBM is fundamentally impossible. Our analysis for the Stochastic Block Model follows the method used in [10] but we analyze a different regime. In [10], two clusters are required to have the equal size (Planted Bisection Model), while in our SBM setup, nature picks the label of each node uniformly at random. Thus in our model only the expectation of the sizes of the two communities are equal.

We now define the Stochastic Block Model, which has two parameters $p$ and $q$.

**Definition 1** (Stochastic Block Model). *Let* $0 < q < p < 1$. *A Stochastic Block Model with parameters* $(p,q)$ *is an undirected graph of* $n$ *nodes with the adjacency matrix* $A$, *where each* $A_{ij} \in \{0,1\}$. *Each node is in one of the two classes* $\{+1,-1\}$. *The distribution of true labels* $Y^* = (y_1^*, \ldots, y_n^*)$ *is uniform, i.e., each label* $y_i^*$ *is assigned to* $+1$ *with probability* $0.5$, *and* $-1$ *with probability* $0.5$.

*The adjacency matrix* $A$ *is distributed as follows: if* $y_i^* = y_j^*$ *then* $A_{ij}$ *is Bernoulli with parameter* $p$; *otherwise* $A_{ij}$ *is Bernoulli with parameter* $q$.

The goal is to recover labels $\hat{Y} = (\hat{y}_1, \ldots, \hat{y}_n)$ that are equal to the true labels $Y^*$, given the observation of $A$. We are interested in the information-theoretic lower bounds. Thus, we define the Markov chain $Y^* \to A \to \hat{Y}$. Using Fano's inequality, we obtain the following results.

**Theorem 1.** *In a Stochastic Block Model with parameters $(p, q)$ with $0 < q < p < 1$, if*

$$\frac{(p-q)^2}{q(1-q)} \leq \frac{2\log 2}{n} - \frac{4\log 2}{n^2},$$

*then we have that for any algorithm that a learner could use for picking $\hat{Y}$, the probability of error $\mathbb{P}(\hat{Y} \neq Y^*)$ is greater than or equal to $\frac{1}{2}$.*

Notice that our result for the Stochastic Block Model is similar to the one in [10]. This means that the method of generating labels does not affect the information-theoretic bound.

## 2.2 Exponential Random Graph Model

Exponential Random Graph Models (ERGMs) are a family of distributions on graphs of the following form: $\mathbb{P}(A) = \exp(\phi(A))/\sum_{A'}\exp(\phi(A'))$, where $\phi : \{0,1\}^{n \times n} \to \mathbb{R}$ is some potential function over graphs. Selecting different potential functions enables ERGMs to model various structures in network graphs, for instance, the potential function can be a sum of functions over edges, triplets, cliques, among other choices [16].

In this section we analyze a special case of the Exponential Random Graph Model as a corollary of our results for the Stochastic Block Model, in which the potential function is defined as a sum of functions over edges. That is, $\phi(A) = \sum_{i,j|A_{ij}=1} \phi_{ij}(y_i, y_j)$, where $\phi_{ij}(y_i, y_j) = \beta y_i y_j$ and $\beta > 0$ is a parameter. Simplifying the expression above, we have $\phi(A) = \sum_{i,j} \beta A_{ij} y_i y_j$. This leads to the following definition.

**Definition 2** (Exponential Random Graph Model). *Let $\beta > 0$. An Exponential Random Graph Model with parameter $\beta$ is an undirected graph of $n$ nodes with the adjacency matrix $A$, where each $A_{ij} \in \{0, 1\}$. Each node is in one of the two classes $\{+1, -1\}$. The distribution of true labels $Y^* = (y_1^*, \ldots, y_n^*)$ is uniform, i.e., each label $y_i^*$ is assigned to $+1$ with probability $0.5$, and $-1$ with probability $0.5$.*

*The adjacency matrix $A$ is distributed as follows: $\mathbb{P}(A|Y) = \exp(\beta \sum_{i<j} A_{ij} y_i y_j)/Z(\beta)$, where $Z(\beta) = \sum_{A' \in \{0,1\}^{n \times n}} \exp(\beta \sum_{i<j} A'_{ij} y_i y_j)$.*

The goal is to recover labels $\hat{Y} = (\hat{y}_1, \ldots, \hat{y}_n)$ that are equal to the true labels $Y^*$, given the observation of $A$. Theorem 1 leads to the following result.

**Corollary 1.** *In an Exponential Random Graph Model with parameter $\beta > 0$, if*

$$2(\cosh \beta - 1) \leq \frac{2\log 2}{n} - \frac{4\log 2}{n^2},$$

*then we have that for any algorithm that a learner could use for picking $\hat{Y}$, the probability of error $\mathbb{P}(\hat{Y} \neq Y^*)$ is greater than or equal to $\frac{1}{2}$.*

## 2.3 Latent Space Model

The Latent Space Model (LSM) was first proposed by [19]. The core assumption of the model is that each node has a low-dimensional latent vector associated with it. The latent vectors of nodes in the same community follow a similar pattern. The connectivity of two nodes in the Latent Space Model is determined by the distance between their corresponding latent vectors. Previous works on the Latent Space Model [30] analyzed asymptotic sample complexity, but did not focus on information-theoretic limits for exact recovery.

We now define the Latent Space Model, which has three parameters $\sigma > 0$, $d \in \mathbb{Z}^+$ and $\mu \in \mathbb{R}^d$, $\mu \neq \mathbf{0}$.

**Definition 3** (Latent Space Model). *Let $d \in \mathbb{Z}^+, \mu \in \mathbb{R}^d$ and $\mu \neq \mathbf{0}, \sigma > 0$. A Latent Space Model with parameters $(d, \mu, \sigma)$ is an undirected graph of $n$ nodes with the adjacency matrix $A$, where each $A_{ij} \in \{0, 1\}$. Each node is in one of the two classes $\{+1, -1\}$. The distribution of true labels $Y^* = (y_1^*, \ldots, y_n^*)$ is uniform, i.e., each label $y_i^*$ is assigned to $+1$ with probability $0.5$, and $-1$ with probability $0.5$.*

*For every node $i$, nature generates a latent $d$-dimensional vector $z_i \in \mathbb{R}^d$ according to the Gaussian distribution $N_d(y_i\mu, \sigma^2\mathbf{I})$.*

*The adjacency matrix $A$ is distributed as follows: $A_{ij}$ is Bernoulli with parameter $\exp(-\|z_i - z_j\|_2^2)$.*

The goal is to recover labels $\hat{Y} = (\hat{y}_1, \ldots, \hat{y}_n)$ that are equal to the true labels $Y^*$, given the observation of $A$. Notice that we do not have access to $Z$. we are interested in the information-theoretic lower bounds. Thus, we define the Markov chain $Y^* \to A \to \hat{Y}$. Fano's inequality and a proper conversion of the above model lead to the following theorem.

**Theorem 2.** *In a Latent Space Model with parameters $(d, \mu, \sigma)$, if*

$$(4\sigma^2 + 1)^{-1-d/2}\|\mu\|_2^2 \leq \frac{\log 2}{2n} - \frac{\log 2}{n^2},$$

*then we have that for any algorithm that a learner could use for picking $\hat{Y}$, the probability of error $\mathbb{P}(\hat{Y} \neq Y^*)$ is greater than or equal to $\frac{1}{2}$.*

## 3 Dynamic Network Models

In this section we analyze the information-theoretic limits for two dynamic network models: the Dynamic Stochastic Block Model (DSBM) and the Dynamic Latent Space Model (DLSM). We call these dynamic models, because we assume there exists some ordering for edges, and the distribution of each edge not only depends on its endpoints, but also depends on previously generated edges.

We start by giving the definition of predecessor sets. Notice that the following definition of predecessor sets employs a lexicographic order, and the motivation is to use it as a subclass to provide a bound for general dynamic models. Fano's inequality is usually used for a restricted ensemble, i.e., a subclass of the original class of interest. If a subclass (e.g., dynamic SBM or LSM with a particular predecessor set $\tau$) is difficult to be learnt, then the original class (SBMs or LSMs with general dynamic interactions) will be at least as difficult to be learnt. The use of restricted ensembles is customary for information-theoretic lower bounds [28, 31].

**Definition 4.** *For every pair $i$ and $j$ with $i < j$, we denote its predecessor set using $\tau_{i,j}$, where*

$$\tau_{ij} \subseteq \{(k,l)|(k < l) \wedge (k < i \vee (k = i \wedge l < j))\}$$

*and*

$$A_{\tau_{ij}} = \{A_{kl}|(k,l) \in \tau_{ij}\}.$$

In a dynamic model, the probability distribution of each edge $A_{ij}$ not only depends on the labels of nodes $i$ and $j$ (i.e., $y_i^*$ and $y_j^*$), but also on the previously generated edges $A_{\tau_{ij}}$.

Next, we prove the following lemma using the definition above.

**Lemma 1.** *Assume now the probability distribution of $A$ given labeling $Y$ is $\mathbb{P}(A|Y) = \prod_{i<j} \mathbb{P}(A_{ij}|A_{\tau_{ij}}, y_i, y_j)$. Then for any labeling $Y$ and $Y'$, we have*

$$\mathbb{KL}(P_{A|Y}\|P_{A|Y'}) \leq \binom{n}{2} \max_{i,j} \mathbb{KL}(P_{A_{ij}|A_{\tau_{ij}}, y_i, y_j}\|P_{A_{ij}|A_{\tau_{ij}}, y_i', y_j'}).$$

*Similarly, if the probability distribution of $A$ given labeling $Y$ is $\mathbb{P}(A|Y) = \prod_{i<j} \mathbb{P}(A_{ij}|A_{\tau_{ij}}, y_1, \ldots, y_j)$, we have*

$$\mathbb{KL}(P_{A|Y}\|P_{A|Y'}) \leq \binom{n}{2} \max_{i,j} \mathbb{KL}(P_{A_{ij}|A_{\tau_{ij}}, y_1, \ldots, y_j}\|P_{A_{ij}|A_{\tau_{ij}}, y_1', \ldots, y_j'}).$$

## 3.1 Dynamic Stochastic Block Model

The Dynamic Stochastic Block Model (DSBM) shares a similar setting with the Stochastic Block Model, except that we take the predecessor sets into consideration.

**Definition 5** (Dynamic Stochastic Block Model). *Let $0 < q < p < 1$. Let $F = \{f_k\}_{k=0}^{\binom{n}{2}}$ be a set of functions, where $f_k : \{0,1\}^k \to (0,1]$. A Dynamic Stochastic Block Model with parameters $(p, q, F)$ is an undirected graph of $n$ nodes with the adjacency matrix $A$, where each $A_{ij} \in \{0,1\}$. Each node is in one of the two classes $\{+1, -1\}$. The distribution of true labels $Y^* = (y_1^*, \ldots, y_n^*)$ is uniform, i.e., each label $y_i^*$ is assigned to $+1$ with probability $0.5$, and $-1$ with probability $0.5$.*

*The adjacency matrix $A$ is distributed as follows: if $y_i^* = y_j^*$ then $A_{ij}$ is Bernoulli with parameter $p f_{|\tau_{ij}|}(A_{\tau_{ij}})$; otherwise $A_{ij}$ is Bernoulli with parameter $q f_{|\tau_{ij}|}(A_{\tau_{ij}})$.*

The goal is to recover labels $\hat{Y} = (\hat{y}_1, \ldots, \hat{y}_n)$ that are equal to the true labels $Y^*$, given the observation of $A$. We are interested in the information-theoretic lower bounds. Thus, we define the Markov chain $Y^* \to A \to \hat{Y}$. Using Fano's inequality and Lemma 1, we obtain the following results.

**Theorem 3.** *In a Dynamic Stochastic Block Model with parameters $(p, q)$ with $0 < q < p < 1$, if*

$$\frac{(p-q)^2}{q(1-q)} \leq \frac{n-2}{n^2-n} \log 2,$$

*then we have that for any algorithm that a learner could use for picking $\hat{Y}$, the probability of error $\mathbb{P}(\hat{Y} \neq Y^*)$ is greater than or equal to $\frac{1}{2}$.*

## 3.2 Dynamic Latent Space Model

The Dynamic Latent Space Model (DLSM) shares a similar setting with the Latent Space Model, except that we take the predecessor sets into consideration.

**Definition 6** (Dynamic Latent Space Model). *Let $d \in \mathbb{Z}^+, \mu \in \mathbb{R}^d$ and $\mu \neq \mathbf{0}, \sigma > 0$. Let $F = \{f_k\}_{k=0}^{\binom{n}{2}}$ be a set of functions, where $f_k : \{0,1\}^k \to (0,1]$. A Latent Space Model with parameters $(d, \mu, \sigma, F)$ is an undirected graph of $n$ nodes with the adjacency matrix $A$, where each $A_{ij} \in \{0,1\}$. Each node is in one of the two classes $\{+1, -1\}$. The distribution of true labels $Y^* = (y_1^*, \ldots, y_n^*)$ is uniform, i.e., each label $y_i^*$ is assigned to $+1$ with probability $0.5$, and $-1$ with probability $0.5$.*

*For every node $i$, nature generates a latent $d$-dimensional vector $z_i \in \mathbb{R}^d$ according to the Gaussian distribution $N_d(y_i\mu, \sigma^2\mathbf{I})$.*

*The adjacency matrix $A$ is distributed as follows: $A_{ij}$ is Bernoulli with parameter $f_{|\tau_{ij}|}(A_{\tau_{ij}}) \cdot \exp(-\|z_i - z_j\|_2^2)$.*

The goal is to recover labels $\hat{Y} = (\hat{y}_1, \ldots, \hat{y}_n)$ that are equal to the true labels $Y^*$, given the observation of $A$. Notice that we do not have access to $Z$. We are interested in the information-theoretic lower bounds. Thus, we define the Markov chain $Y^* \to A \to \hat{Y}$. Using Fano's inequality and Lemma 1, our analysis leads to the following theorem.

**Theorem 4.** *In a Dynamic Latent Space Model with parameters $(d, \mu, \sigma, \{f_k\})$, if*

$$(4\sigma^2 + 1)^{-1-d/2}\|\mu\|_2^2 \leq \frac{n-2}{4(n^2-n)} \log 2,$$

*then we have that for any algorithm that a learner could use for picking $\hat{Y}$, the probability of error $\mathbb{P}(\hat{Y} \neq Y^*)$ is greater than or equal to $\frac{1}{2}$.*

# 4 Directed Network Models

In this section we analyze the information-theoretic limits for two directed network models: the Directed Preferential Attachment Model (DPAM) and the Directed Small-world Model (DSWM). In contrast to previous sections, here we consider directed graphs.

Note that in social networks such as Twitter, the graph is directed. That is, each user follows other users. Users that are followed by many others (i.e., nodes with high out-degree) are more likely to be followed by new users. This is the case of popular singers, for instance. Additionally, a new user will follow people with similar preferences. This is referred in the literature as homophily. In our case, a node with positive label will more likely follow nodes with positive label, and vice versa.

The two models defined in this section will require an expected number of in-neighbors $m$, for each node. In order to guarantee this in a setting in which nodes decide to connect to at most $k > m$ nodes independently, one should guarantee that the probability of choosing each of the $k$ nodes is less than or equal to $1/m$.

The above motivates an algorithm that takes a vector in the $k$-simplex (i.e., $w \in R^k$ and $\sum_{i=1}^{k} w_i = 1$) and produces another vector in the $k$-simplex (i.e., $\tilde{w} \in R^k$, $\sum_{i=1}^{k} \tilde{w}_i = 1$ and for all $i$, $\tilde{w}_i \leq 1/m$). Consider the following optimization problem:

$$\operatorname*{minimize}_{\tilde{w}} \quad \frac{1}{2} \sum_{i=1}^{k} (\tilde{w}_i - w_i)^2$$

$$\text{subject to} \quad 0 \leq \tilde{w}_i \leq \frac{1}{m} \text{ for all } i$$

$$\sum_{i=1}^{k} \tilde{w}_i = 1.$$

which is solved by the following algorithm:

---
**Algorithm 1:** $k$-simplex

---
**input** : vector $w \in \mathbb{R}^k$ where $\sum_{i=1}^{k} w_i = 1$,
        expected number of in-neighbors $m \leq k$
**output** : vector $\tilde{w} \in \mathbb{R}^k$ where $\sum_{i=1}^{k} \tilde{w}_i = 1$ and $\tilde{w}_i \leq 1/m$ for all $i$
1 **for** $i \in \{1, \ldots, k\}$ **do**
2   |   $\tilde{w}_i \leftarrow w_i$;
3 **end**
4 **for** $i \in \{1, \ldots, k\}$ such that $\tilde{w}_i > \frac{1}{m}$ **do**
5   |   $S \leftarrow \tilde{w}_i - \frac{1}{m}$;
6   |   $\tilde{w}_i \leftarrow \frac{1}{m}$;
7   |   Distribute $S$ evenly across all $j \in \{1, \ldots, k\}$ such that $\tilde{w}_j < \frac{1}{m}$;
8 **end**

---

One important property that we will use in our proofs is that $\min_i \tilde{w}_i \geq \min_i w_i$, as well as $\max_i \tilde{w}_i \leq \max_i w_i$.

## 4.1 Directed Preferential Attachment Model

Here we consider a Directed Preferential Attachment Model (DPAM) based on the classic Preferential Attachment Model [7]. While in the classic model every mode has exactly $m$ neighbors, in our model the expected number of in-neighbors is $m$.

**Definition 7** (Directed Preferential Attachment Model). *Let $m$ be a positive integer with $0 < m \ll n$. Let $s > 0$ be the homophily parameter. A Directed Preferential Attachment Model with parameters $(m, s)$ is a directed graph of $n$ nodes with the adjacency matrix $A$, where each $A_{ij} \in \{0, 1\}$. Each node is in one of the two classes $\{+1, -1\}$. The distribution of true labels $Y^* = (y_1^*, \ldots, y_n^*)$ is uniform, i.e., each label $y_i^*$ is assigned to $+1$ with probability $0.5$, and $-1$ with probability $0.5$.*

*Nodes $1$ through $m$ are not connected to each other, and they all have an in-degree of $0$. For node $i$ from $m + 1$ to $n$, nature first generates the weight $w_{ji}$ for each node $j < i$, where $w_{ji} \propto$*

$(\sum_{k=1}^{i-1} A_{jk} + 1)(\mathbf{1}[y_i^* = y_j^*]s + 1)$, and $\sum_{j=1}^{i-1} w_{ji} = 1$. Then every node $j < i$ connects to node $i$ with the following probability: $\mathbb{P}(A_{ji} = 1 \mid A_{\tau_{ij}}, y_1^*, \ldots, y_j^*) = m\tilde{w}_{ji}$, where $(\tilde{w}_{1i}...\tilde{w}_{i-1,i})$ is computed from $(w_{1i}...w_{i-1,i})$ as in Algorithm 1.

The goal is to recover labels $\hat{Y} = (\hat{y}_1, \ldots, \hat{y}_n)$ that are equal to the true labels $Y^*$, given the observation of $A$. We are interested in the information-theoretic lower bounds. Thus, we define the Markov chain $Y^* \to A \to \hat{Y}$. Using Fano's inequality, we obtain the following results.

**Theorem 5.** *In a Directed Preferential Attachment Model with parameters* $(m, s)$, *if*

$$\frac{s+1}{8m} \leq \frac{2^{(n-2)/(n^2-n)}}{n^2},$$

*then we have that for any algorithm that a learner could use for picking $\hat{Y}$, the probability of error $\mathbb{P}(\hat{Y} \neq Y^*)$ is greater than or equal to $\frac{1}{2}$.*

## 4.2 Directed Small-world Model

Here we consider a Directed Small-world Model (DSWM) based on the classic small-world phenomenon [32]. While in the classic model every mode has exactly $m$ neighbors, in our model the expected number of in-neighbors is $m$.

**Definition 8** (Directed Small-world Model). *Let $m$ be a positive integer with $0 < m \ll n$. Let $s > 0$ be the homophily parameter. Let $p$ be the mixture parameter with $0 < p < 1$. A Directed Small-world Model with parameters $(m, s, p)$ is a directed graph of $n$ nodes with the adjacency matrix $A$, where each $A_{ij} \in \{0, 1\}$. Each node is in one of the two classes $\{+1, -1\}$. The distribution of true labels $Y^* = (y_1^*, \ldots, y_n^*)$ is uniform, i.e., each label $y_i^*$ is assigned to $+1$ with probability $0.5$, and $-1$ with probability $0.5$.*

*Nodes $1$ through $m$ are not connected to each other, and they all have an in-degree of $0$. For node $i$ from $m + 1$ to $n$, nature first generates the weight $w_{ji}$ for each node $j < i$, where $w_{ji} \propto (\mathbf{1}[y_i^* = y_j^*]s + 1)$, and $\sum_{j=i-m}^{i-1} w_{ji} = p$, $\sum_{j=1}^{i-m-1} w_{ji} = 1 - p$. Then every node $j < i$ connects to node $i$ with the following probability: $\mathbb{P}(A_{ji} = 1 \mid A_{\tau_{ij}}, y_1^*, \ldots, y_j^*) = m\tilde{w}_{ji}$, where $(\tilde{w}_{1i}...\tilde{w}_{i-1,i})$ is computed from $(w_{1i}...w_{i-1,i})$ as in Algorithm 1.*

The goal is to recover labels $\hat{Y} = (\hat{y}_1, \ldots, \hat{y}_n)$ that are equal to the true labels $Y^*$, given the observation of $A$. We are interested in the information-theoretic lower bounds. Thus, we define the Markov chain $Y^* \to A \to \hat{Y}$. Using Fano's inequality, we obtain the following results.

**Theorem 6.** *In a Directed Small-world Model with parameters $(m, s, p)$, if*

$$\frac{(s+1)^2}{mp(1-p)} \leq \frac{2^{2(n-2)/n^2}}{n},$$

*then we have that for any algorithm that a learner could use for picking $\hat{Y}$, the probability of error $\mathbb{P}(\hat{Y} \neq Y^*)$ is greater than or equal to $\frac{1}{2}$.*

## 5 Concluding Remarks

In the past decade a lot of effort has been made in the Stochastic Block Model (SBM) community to find polynomial time algorithms for the exact recovery. For example, [2, 10] provided analyses to various parameter regimes in symmetric SBMs, and showed that some easy regimes could be solved in polynomial time using semidefinite programming relaxation; [3] also provides quasi-linear time algorithms for SBMs; [17] and [6] discovered the existence of phase transition in the exact recovery of symmetric SBMs. All of the aforementioned literature has mathematical guarantees of statistical and computational efficiency. There exists algorithms without formal guarantees, for example, [16] introduced some MCMC-based methods. Other heuristic algorithms include Kernighan-Lin's algorithm, METIS, Local Spectral Partitioning, etc. (See e.g., [22] for reference.)

We want to highlight that community detection for undirected models could be viewed as a special case of the Markov random field (MRF) inference problem. In the MRF model, if the pairwise potentials are submodular, the problem could be solved exactly in polynomial time via graph cuts in the case of two communities [8].

Regarding our contributions, we highlight that the entries in the adjacency matrix $A$ are not independent in several models considered in our paper, including the Dynamic Stochastic Block Model, the Dynamic Latent Space Model, the Directed Preferential Attachment Model and the Directed Small-world Model. Also, in the Latent Space Model and the Dynamic Latent Space Model, we have additional latent variables. Furthermore, in the Directed Preferential Attachment Model and the Directed Small-world Model, an entry in $A$ also depends on several entries in $Y^*$ to account for homophily.

Our research could be extended in several ways. First, our models only involve two symmetric clusters. For the Latent Space Model and dynamic models, it might be interesting to analyze the case with multiple clusters. Some more complicated models involving Markovian assumptions, for example, the Dynamic Social Network in Latent Space model [29], can also be analyzed. We acknowledge that the information-theoretic lower bounds we provide in this paper may not be necessarily tight. It would be interesting to analyze phase transitions and information-computational gaps for the new models.

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
