[Supplementary Material]

# A Static Network Models

## A.1 Proof of Theorem 1

*Proof.* We use $\mathcal{Y}$ to denote the hypothesis class, which has the size of $|\mathcal{Y}| = 2^n$. By Fano's inequality [12], we have for any $\hat{Y}$,

$$
\begin{aligned}
\mathbb{P}(\hat{Y} \neq Y^*) &\geq 1 - \frac{I(Y^*, A) + \log 2}{\log |\mathcal{Y}|} \\
&= 1 - \frac{I(Y^*, A) + \log 2}{n \log 2}.
\end{aligned} \tag{1}
$$

Our main step is to give an upper bound for the mutual information $I(Y^*, A)$ in order to apply Fano's inequality. By using the pairwise KL-based bound from [35, p. 428] we have

$$
\begin{aligned}
I(Y^*, A) &\leq \frac{1}{|\mathcal{Y}|^2} \sum_{Y \in \mathcal{Y}} \sum_{Y' \in \mathcal{Y}} \mathbb{KL}(P_{A|Y} \| P_{A|Y'}) \\
&\leq \max_{Y, Y' \in \mathcal{Y}} \mathbb{KL}(P_{A|Y} \| P_{A|Y'}) \\
&= \max_{Y, Y' \in \mathcal{Y}} \sum_A \mathbb{P}(A|Y) \log \frac{\mathbb{P}(A|Y)}{\mathbb{P}(A|Y')} \\
&\leq^{(a)} \frac{n^2}{4} \max_{y_i, y_j, y_i', y_j'} \sum_{A_{ij}} \mathbb{P}(A_{ij}|y_i, y_j) \log \frac{\mathbb{P}(A_{ij}|y_i, y_j)}{\mathbb{P}(A_{ij}|y_i', y_j')} \\
&=^{(b)} \frac{n^2}{4} \cdot \sum_{A_{ij}} \mathbb{P}(A_{ij}|y_i = y_j) \log \frac{\mathbb{P}(A_{ij}|y_i = y_j)}{\mathbb{P}(A_{ij}|y_i \neq y_j)} \\
&= \frac{n^2}{4} \cdot \left( p \log \frac{p}{q} + (1-p) \log \frac{1-p}{1-q} \right) \\
&= \frac{n^2}{4} \cdot \mathbb{KL}(p \| q).
\end{aligned} \tag{2}
$$

Among the equations above, (a) holds because $A$ is symmetric, and $A_{ij}$'s are independent and identically distributed given $Y$, while (b) holds because for every $i$ and $j$, we have

$$
\sum_{A_{ij}} \mathbb{P}(A_{ij}|y_i = y_j) \log \frac{\mathbb{P}(A_{ij}|y_i = y_j)}{\mathbb{P}(A_{ij}|y_i \neq y_j)} > \sum_{A_{ij}} \mathbb{P}(A_{ij}|y_i \neq y_j) \log \frac{\mathbb{P}(A_{ij}|y_i \neq y_j)}{\mathbb{P}(A_{ij}|y_i = y_j)},
$$

given that $p > q$. Next we use formula (16) from [10]:

$$
\begin{aligned}
\mathbb{KL}(p \| q) &= \left( p \log \frac{p}{q} + (1-p) \log \frac{1-p}{1-q} \right) \\
&\leq p \frac{p-q}{q} + (1-p) \frac{q-p}{1-q} \\
&= \frac{(p-q)^2}{q(1-q)}.
\end{aligned} \tag{3}
$$

By Fano's inequality [12] and by plugging (3) and (2) into (1), for the probability error to be at least $1/2$, it is sufficient for the lower bound to be greater than $1/2$. Therefore

$$
\mathbb{P}(\hat{Y} \neq \bar{Y}) \geq 1 - \frac{I(Y^*, A) + \log 2}{n \log 2} \geq 1 - \frac{\frac{n^2}{4} \cdot \frac{(p-q)^2}{q(1-q)} + \log 2}{n \log 2} \geq \frac{1}{2}.
$$

By solving for $n$ in the inequality above, we obtain that if

$$
\frac{(p-q)^2}{q(1-q)} \leq \frac{2 \log 2}{n} - \frac{4 \log 2}{n^2}, \tag{4}
$$

then we have that $\mathbb{P}(\hat{Y} \neq \bar{Y}) \geq \frac{1}{2}$. $\qquad\square$

## A.2 Proof of Corollary 1

*Proof.* Starting from the probability distribution of the adjacency matrix $A$, we have

$$
\begin{aligned}
\mathbb{P}(A|Y) &= \frac{\exp(\beta \sum_{i<j} A_{ij} y_i y_j)}{\sum_{A' \in \{0,1\}^{n \times n}} \exp(\beta \sum_{i<j} A'_{ij} y_i y_j)} \\
&= \frac{\prod_{i<j} \exp(\beta A_{ij} y_i y_j)}{\prod_{i<j}(1 + \exp(\beta y_i y_j))} \\
&= \prod_{i<j} \frac{\exp(\beta A_{ij} y_i y_j)}{1 + \exp(\beta y_i y_j)} \\
&= \prod_{i<j} \mathbb{P}(A_{ij} | y_i, y_j).
\end{aligned}
$$

Thus, $A_{ij}$ is Bernoulli with parameter $\frac{\exp(\beta A_{ij} y_i y_j)}{1+\exp(\beta y_i y_j)}$. We denote $p = \mathbb{P}(A_{ij}|y_i = y_j) = \frac{\exp(\beta)}{1+\exp(\beta)}$, and $q = \mathbb{P}(A_{ij}|y_i \neq y_j) = \frac{\exp(-\beta)}{1+\exp(-\beta)}$. Plugging $p$ and $q$ into (4) and requiring the probability error to be at least $1/2$, we obtain that if

$$
2(\cosh \beta - 1) \leq \frac{2 \log 2}{n} - \frac{4 \log 2}{n^2}, \tag{5}
$$

then we have that $\mathbb{P}(\hat{Y} \neq \bar{Y}) \geq \frac{1}{2}$. $\qquad\square$

## A.3 Moment Generating Function of Multivariate Gaussian Distribution

We introduce the following result from [24, p. 40], which will later be used in the proof of Theorem 2 and 4.

**Lemma 2.** *Let $x \sim N_p(\mu, \Sigma)$, $Q = x^\top A x$, $A = A^\top$. Then the moment generating function of $Q$ is given by*

$$
\begin{aligned}
M_Q(t) &= \mathbb{E}_{x \sim N_p(\mu,\Sigma)}[\exp(t x^\top A x)] \\
&= \int_x \frac{\exp(t x^\top A x - \frac{1}{2}(x - \mu)^\top \Sigma^{-1}(x - \mu)^\top)}{(2\pi)^{p/2} |\Sigma|^{1/2}} dx.
\end{aligned}
$$

*Furthermore, if $(\Sigma^{-1} - 2tA)$ is symmetric positive definite, we have*

$$
\begin{aligned}
M_Q(t) =& |I - 2t\Sigma^{1/2} A \Sigma^{1/2}|^{-1/2} \\
& \cdot \exp\left(t\mu^\top \Sigma^{-1/2}(\Sigma^{1/2} A \Sigma^{1/2}) \cdot (I - 2t\Sigma^{1/2} A \Sigma^{1/2})^{-1} \Sigma^{-1/2}\mu\right).
\end{aligned}
$$

## A.4 Proof of Theorem 2

First, we start with a required technical lemma:

**Lemma 3.** *The model considered in Definition 3 is equivalent to the following Modified Latent Space Model:*

*Let $d \in \mathbb{Z}^+, \mu \in \mathbb{R}^d$ and $\mu \neq 0, \sigma > 0$. A modified Latent Space Model with parameters $(d, \mu, \sigma)$ is an undirected graph of $n$ nodes with the adjacency matrix $A$, where each $A_{ij} \in \{0,1\}$. Each node is in one of the two classes $\{+1, -1\}$. The distribution of true labels $Y^* = (y_1^*, \ldots, y_n^*)$ is uniform, i.e., each label $y_i^*$ is assigned to $+1$ with probability $0.5$, and $-1$ with probability $0.5$.*

*For every node $i$, the nature generates a latent $d$-dimensional vector $x_i \in \mathbb{R}^d$ according to the Gaussian distribution $N_d(\mathbf{0}, \sigma^2 \mathbf{I})$.*

*The adjacency matrix $A$ is distributed as follows: if $y_i^* = y_j^*$ then $A_{ij}$ is Bernoulli with parameter $\exp(-\|x_i - x_j\|_2^2)$; otherwise $A_{ij}$ is Bernoulli with parameter $\exp(-\|x_i - x_j + 2y_i^*\mu\|_2^2)$.*

*Proof.* We claim that the Modified Latent Space Model is equivalent to the classic Latent Space Model considered in Definition 3, by defining $x_i = z_i - y_i\mu$ for every node $i$. Since $z_i \sim N_d(y_i\mu, \sigma^2\mathbf{I})$, we have $x_i \sim N_d(\mathbf{0}, \sigma^2\mathbf{I})$. As a result,

- if $y_i^* = y_j^*$, $A_{ij}$ is Bernoulli with parameter $\exp(-\|z_i - z_j\|_2^2) = \exp(-\|x_i + y_i^*\mu - x_j - y_j^*\mu\|_2^2) = \exp(-\|x_i - x_j\|_2^2)$,

- if $y_i^* = 1, y_j^* = -1$, $A_{ij}$ is Bernoulli with parameter $\exp(-\|z_i - z_j\|_2^2) = \exp(-\|x_i + \mu - x_j + \mu\|_2^2) = \exp(-\|x_i - x_j + 2\mu\|_2^2)$,

- if $y_i^* = -1, y_j^* = 1$, $A_{ij}$ is Bernoulli with parameter $\exp(-\|z_i - z_j\|_2^2) = \exp(-\|x_i - \mu - x_j - \mu\|_2^2) = \exp(-\|x_i - x_j - 2\mu\|_2^2)$.

This completes the proof of the lemma. $\qquad\square$

Now, we provide the proof of the main theorem.

*Proof.* Since $X$ and $Y$ are independent, we have the following equalities

$$
\begin{aligned}
\mathbb{P}(A_{ij}|y_i, y_j) &= \int_{x_i, x_j} \mathbb{P}(A_{ij}, x_i, x_j|y_i, y_j)dx_i dx_j \\
&= \int_{x_i, x_j} \mathbb{P}(x_i, x_j|y_i, y_j)\mathbb{P}(A_{ij}|y_i, y_j, x_i, x_j)dx_i dx_j \\
&= \int_{x_i, x_j} \mathbb{P}(x_i, x_j)\mathbb{P}(A_{ij}|y_i, y_j, x_i, x_j)dx_i dx_j \\
&= \mathbb{E}_{x_i, x_j}[\mathbb{P}(A_{ij}|y_i, y_j, x_i, x_j)].
\end{aligned}
\tag{6}
$$

The second last equality holds because in the modified model, latent vectors are independent of their labels. Now we are interested in the expectations $\mathbb{E}_{x_i, x_j}[\mathbb{P}(A_{ij} = 1|y_i = y_j, x_i, x_j)]$ and $\mathbb{E}_{x_i, x_j}[\mathbb{P}(A_{ij} = 1|y_i \neq y_j, x_i, x_j)]$. By definition we know

$$
\mathbb{E}_{x_i, x_j}[\mathbb{P}(A_{ij} = 1|y_i = y_j, x_i, x_j)] = \mathbb{E}_{x_i, x_j}[\exp(-\|x_i - x_j\|_2^2)],
$$

and

$$
\begin{aligned}
&\mathbb{E}_{x_i, x_j}[\mathbb{P}(A_{ij} = 1|y_i \neq y_j, x_i, x_j)] \\
&= \mathbb{P}(y_i = 1, y_j = -1|y_i \neq y_j) \cdot \mathbb{E}_{x_i, x_j}[\mathbb{P}(A_{ij} = 1|y_i = 1, y_j = -1, x_i, x_j)] \\
&\quad + \mathbb{P}(y_i = -1, y_j = 1|y_i \neq y_j) \cdot \mathbb{E}_{x_i, x_j}[\mathbb{P}(A_{ij} = 1|y_i = -1, y_j = 1, x_i, x_j)] \\
&= \frac{1}{2}\Big(\mathbb{E}_{x_i, x_j}[\mathbb{P}(A_{ij} = 1|y_i = 1, y_j = -1, x_i, x_j)] + \mathbb{E}_{x_i, x_j}[\mathbb{P}(A_{ij} = 1|y_i = -1, y_j = 1, x_i, x_j)]\Big).
\end{aligned}
$$

Since $x_i, x_j$ follow the distribution $N_d(\mathbf{0}, \sigma^2\mathbf{I})$, we have $x_i - x_j \sim N_d(\mathbf{0}, 2\sigma^2\mathbf{I})$, $x_i - x_j + 2y_i\mu \sim N_d(2y_i\mu, 2\sigma^2\mathbf{I})$. Thus we can use Lemma 2 in Appendix A.3 with $t = -1$ and obtain the following results:

$$
\begin{aligned}
\mathbb{E}_{x_i, x_j}[\mathbb{P}(A_{ij} = 1|y_i = y_j, x_i, x_j)] &= (4\sigma^2 + 1)^{-d/2} \\
\mathbb{E}_{x_i, x_j}[\mathbb{P}(A_{ij} = 1|y_i = 1, y_j = -1, x_i, x_j)] &= (4\sigma^2 + 1)^{-d/2} \cdot \exp(-\frac{4\|\mu\|_2^2}{4\sigma^2 + 1}) \\
\mathbb{E}_{x_i, x_j}[\mathbb{P}(A_{ij} = 1|y_i = -1, y_j = 1, x_i, x_j)] &= (4\sigma^2 + 1)^{-d/2} \cdot \exp(-\frac{4\|\mu\|_2^2}{4\sigma^2 + 1}).
\end{aligned}
\tag{7}
$$

Notice that $0 < \mathbb{E}_{x_i,x_j}[\mathbb{P}(A_{ij} = 1|y_i \neq y_j, x_i, x_j)] < \mathbb{E}_{x_i,x_j}[\mathbb{P}(A_{ij} = 1|y_i = y_j, x_i, x_j)] < 1$. By using the pairwise KL-based bound from [35, p. 428] we have

$$
\begin{aligned}
I(Y^*, A) &\leq \frac{1}{|\mathcal{Y}|^2} \sum_{Y \in \mathcal{Y}} \sum_{Y' \in \mathcal{Y}} \mathbb{KL}(P_{A|Y} \| P_{A|Y'}) \\
&\leq \max_{Y,Y' \in \mathcal{Y}} \mathbb{KL}(P_{A|Y} \| P_{A|Y'}) \\
&= \max_{Y,Y' \in \mathcal{Y}} \sum_A \mathbb{P}(A|Y) \log \frac{\mathbb{P}(A|Y)}{\mathbb{P}(A|Y')} \\
&\leq \frac{n^2}{4} \max_{y_i,y_j,y_i',y_j'} \sum_{A_{ij}} \mathbb{P}(A_{ij}|y_i, y_j) \log \frac{\mathbb{P}(A_{ij}|y_i, y_j)}{\mathbb{P}(A_{ij}|y_i', y_j')} \\
&= \frac{n^2}{4} \max_{y_i,y_j,y_i',y_j'} \sum_{A_{ij}} \mathbb{E}_{x_i,x_j}[\mathbb{P}(A_{ij}|y_i, y_j, x_i, x_j)] \cdot \log \frac{\mathbb{E}_{x_i,x_j}[\mathbb{P}(A_{ij}|y_i, y_j, x_i, x_j)]}{\mathbb{E}_{x_i,x_j}[\mathbb{P}(A_{ij}|y_i', y_j', x_i, x_j)]} \\
&= \sum_{A_{ij}} \mathbb{E}_{x_i,x_j}[\mathbb{P}(A_{ij}|y_i = y_j, x_i, x_j)] \cdot \log \frac{\mathbb{E}_{x_i,x_j}[\mathbb{P}(A_{ij}|y_i = y_j, x_i, x_j)]}{\mathbb{E}_{x_i,x_j}[\mathbb{P}(A_{ij}|y_i \neq y_j, x_i, x_j)]} \\
&<^{(c)} \mathbb{E}_{x_i,x_j}[\mathbb{P}(A_{ij} = 1|y_i = y_j, x_i, x_j)] \cdot \log \frac{\mathbb{E}_{x_i,x_j}[\mathbb{P}(A_{ij} = 1|y_i = y_j, x_i, x_j)]}{\mathbb{E}_{x_i,x_j}[\mathbb{P}(A_{ij} = 1|y_i \neq y_j, x_i, x_j)]} \\
&= n^2(4\sigma^2 + 1)^{-1-d/2}\|\mu\|_2^2,
\end{aligned}
\tag{8}
$$

where (c) holds because for every $i$ and $j$, we have

$$
\begin{aligned}
&\mathbb{E}_{x_i,x_j}[\mathbb{P}(A_{ij} = 0|y_i = y_j, x_i, x_j)] \cdot \log \frac{\mathbb{E}_{x_i,x_j}[\mathbb{P}(A_{ij} = 0|y_i = y_j, x_i, x_j)]}{\mathbb{E}_{x_i,x_j}[\mathbb{P}(A_{ij} = 0|y_i \neq y_j, x_i, x_j)]} \\
&= (1 - \mathbb{E}_{x_i,x_j}[\mathbb{P}(A_{ij} = 1|y_i = y_j, x_i, x_j)]) \cdot \log \frac{1 - \mathbb{E}_{x_i,x_j}[\mathbb{P}(A_{ij} = 1|y_i = y_j, x_i, x_j)]}{1 - \mathbb{E}_{x_i,x_j}[\mathbb{P}(A_{ij} = 1|y_i \neq y_j, x_i, x_j)]} \\
&= (1 - (4\sigma^2 + 1)^{-d/2}) \cdot \log \frac{1 - (4\sigma^2 + 1)^{-p/2}}{1 - (4\sigma^2 + 1)^{-d/2} \cdot \exp(-\frac{4\|\mu\|_2^2}{4\sigma^2+1})} \\
&< 0.
\end{aligned}
$$

Thus, we only need to consider the case for $A_{ij} = 1$.

By Fano's inequality [12] and by plugging the result (8) into (1), for the probability error to be at least $1/2$, it is sufficient for the lower bound to be greater than $1/2$. Therefore we obtain that if

$$
(4\sigma^2 + 1)^{-1-d/2}\|\mu\|_2^2 \leq \frac{\log 2}{2n} - \frac{\log 2}{n^2},
\tag{9}
$$

then for any estimator $\hat{Y}$, $\mathbb{P}(\hat{Y} \neq Y^*) \geq \frac{1}{2}$. $\qquad\square$

# B Dynamic Network Models

## B.1 Proof of Lemma 1

*Proof.* For the first part, starting from the left-hand side, we have

$$
\begin{aligned}
\mathbb{KL}(P_{A|Y}\|P_{A|Y'}) &= \sum_A \mathbb{P}(A|Y) \log \frac{\mathbb{P}(A|Y)}{\mathbb{P}(A|Y')} \\
&= \sum_A \left( \prod_{i<j} \mathbb{P}(A_{ij}|A_{\tau_{ij}}, y_i, y_j) \cdot \log \frac{\prod_{k<l} \mathbb{P}(A_{kl}|A_{\tau_{kl}}, y_k, y_l)}{\prod_{k<l} \mathbb{P}(A_{kl}|A_{\tau_{kl}}, y'_k, y'_l)} \right) \\
&= \sum_A \left( \prod_{i<j} \mathbb{P}(A_{ij}|A_{\tau_{ij}}, y_i, y_j) \cdot \sum_{k<l} \log \frac{\mathbb{P}(A_{kl}|A_{\tau_{kl}}, y_k, y_l)}{\mathbb{P}(A_{kl}|A_{\tau_{kl}}, y'_k, y'_l)} \right) \\
&= \sum_{k<l} \sum_A \left( \prod_{i<j} \mathbb{P}(A_{ij}|A_{\tau_{ij}}, y_i, y_j) \cdot \log \frac{\mathbb{P}(A_{kl}|A_{\tau_{kl}}, y_k, y_l)}{\mathbb{P}(A_{kl}|A_{\tau_{kl}}, y'_k, y'_l)} \right) \\
&= \sum_{k<l} \sum_A \left( P(A_{kl}|A_{\tau_{kl}}, y_k, y_l) \cdot \log \frac{\mathbb{P}(A_{kl}|A_{\tau_{kl}}, y_k, y_l)}{\mathbb{P}(A_{kl}|A_{\tau_{kl}}, y'_k, y'_l)} \right) \\
&= \sum_{i<j} \mathbb{KL}(P_{A_{ij}|A_{\tau_{ij}}, y_i, y_j}\|P_{A_{ij}|A_{\tau_{ij}}, y'_i, y'_j}) \\
&\leq \binom{n}{2} \max_{i,j} \mathbb{KL}(P_{A_{ij}|A_{\tau_{ij}}, y_i, y_j}\|P_{A_{ij}|A_{\tau_{ij}}, y'_i, y'_j}). \tag{10}
\end{aligned}
$$

The proof for the second part follows the same approach. $\square$

## B.2 Proof of Theorem 3

*Proof.* For simplicity we use the shorthand notation $f_{ij} = f_{|\tau_{ij}|}(A_{\tau_{ij}})$. By using the pairwise KL-based bound from [35, p. 428] and Lemma 1, we have

$$
\begin{aligned}
I(Y^*, A) &\leq \frac{1}{|\mathcal{Y}|^2} \sum_{Y \in \mathcal{Y}} \sum_{Y' \in \mathcal{Y}} \mathbb{KL}(P_{A|Y}\|P_{A|Y'}) \\
&\leq \max_{Y,Y' \in \mathcal{Y}} \mathbb{KL}(P_{A|Y}\|P_{A|Y'}) \\
&\leq \max_{y_i, y_j, y'_i, y'_j} \binom{n}{2} \max_{i,j} \mathbb{KL}(P_{A_{ij}|A_{\tau_{ij}}, y_i, y_j}\|P_{A_{ij}|A_{\tau_{ij}}, y'_i, y'_j}) \\
&= \binom{n}{2} \max_{i,j} \sum_{A_{ij}} \mathbb{P}(A_{ij}|A_{\tau_{ij}}, y_i = y_j) \cdot \log \frac{\mathbb{P}(A_{ij}|A_{\tau_{ij}}, y_i = y_j)}{\mathbb{P}(A_{ij}|A_{\tau_{ij}}, y'_i \neq y'_j)} \\
&= \binom{n}{2} \max_{i,j} \left( pf_{ij} \log \frac{pf_{ij}}{qf_{ij}} + (1 - pf_{ij}) \log \frac{1 - pf_{ij}}{1 - qf_{ij}} \right) \\
&= \binom{n}{2} \max_{i,j} \mathbb{KL}(pf_{ij}\|qf_{ij}) \\
&\leq \binom{n}{2} \max_{i,j} pf_{ij} \frac{pf_{ij} - qf_{ij}}{qf_{ij}} + (1 - pf_{ij}) \frac{qf_{ij} - pf_{ij}}{1 - qf_{ij}} \\
&= \binom{n}{2} \max_{i,j} \frac{f_{ij}(p - q)^2}{q(1 - qf_{ij})} \\
&\leq \binom{n}{2} \frac{(p - q)^2}{q(1 - q)}. \tag{11}
\end{aligned}
$$

By Fano's inequality [12] and by plugging (11) into (1), for the probability error to be at least $1/2$, it is sufficient for the lower bound to be greater than 1/2. Therefore

$$\mathbb{P}(\hat{Y} \neq \bar{Y}) \geq 1 - \frac{I(Y^*, A) + \log 2}{n \log 2} \geq 1 - \frac{\frac{n^2 - n}{2} \cdot \frac{(p-q)^2}{q(1-q)} + \log 2}{n \log 2} \geq \frac{1}{2}$$

By solving for $n$ in the inequality above, we obtain that if

$$\frac{(p-q)^2}{q(1-q)} \leq \frac{n-2}{n^2 - n} \log 2, \tag{12}$$

then we have that $\mathbb{P}(\hat{Y} \neq \bar{Y}) \geq \frac{1}{2}$. $\qquad\square$

### B.3 Proof of Theorem 4

First, we start with a required technical lemma:

**Lemma 4.** *The model considered in Definition 6 is equivalent to the following Modified Dynamic Latent Space Model:*

*Let $d \in \mathbb{Z}^+, \mu \in \mathbb{R}^d$ and $\mu \neq 0, \sigma > 0$. Let $F = \{f_k\}_{k=0}^{\binom{n}{2}}$ be a set of functions, where $f_k : \{0,1\}^k \to (0,1]$. A modified Latent Space Model with parameters $(d, \mu, \sigma, F)$ is an undirected graph of $n$ nodes with the adjacency matrix $A$, where each $A_{ij} \in \{0,1\}$. Each node is in one of the two classes $\{+1, -1\}$. The distribution of true labels $Y^* = (y_1^*, \ldots, y_n^*)$ is uniform, i.e., each label $y_i^*$ is assigned to $+1$ with probability $0.5$, and $-1$ with probability $0.5$.*

*For every node $i$, the nature generates a latent $d$-dimensional vector $x_i \in \mathbb{R}^d$ according to the Gaussian distribution $N_d(\mathbf{0}, \sigma^2 \mathbf{I})$.*

*The adjacency matrix $A$ is distributed as follows: if $y_i^* = y_j^*$ then $A_{ij}$ is Bernoulli with parameter $f_{|\tau_{ij}|}(A_{\tau_{ij}}) \cdot \exp(-\|x_i - x_j\|_2^2)$; otherwise $A_{ij}$ is Bernoulli with parameter $f_{|\tau_{ij}|}(A_{\tau_{ij}}) \cdot \exp(-\|x_i - x_j + 2y_i^* \mu\|_2^2)$.*

*Proof.* We claim that the Modified Dynamic Latent Space Model is equivalent to the model considered in Definition 6, by defining $x_i = z_i - y_i \mu$ for every node $i$. Since $z_i \sim N_d(y_i \mu, \sigma^2 \mathbf{I})$, we have $x_i \sim N_d(\mathbf{0}, \sigma^2 \mathbf{I})$. As a result,

- if $y_i^* = y_j^*$, $A_{ij}$ is Bernoulli with parameter $f_{|\tau_{ij}|}(A_{\tau_{ij}}) \cdot \exp(-\|z_i - z_j\|_2^2) = f_{|\tau_{ij}|}(A_{\tau_{ij}}) \cdot \exp(-\|x_i + y_i^* \mu - x_j - y_j^* \mu\|_2^2) = f_{|\tau_{ij}|}(A_{\tau_{ij}}) \cdot \exp(-\|x_i - x_j\|_2^2)$,

- if $y_i^* = 1, y_j^* = -1$, $A_{ij}$ is Bernoulli with parameter $f_{|\tau_{ij}|}(A_{\tau_{ij}}) \cdot \exp(-\|z_i - z_j\|_2^2) = f_{|\tau_{ij}|}(A_{\tau_{ij}}) \cdot \exp(-\|x_i + \mu - x_j + \mu\|_2^2) = f_{|\tau_{ij}|}(A_{\tau_{ij}}) \cdot \exp(-\|x_i - x_j + 2\mu\|_2^2)$,

- if $y_i^* = -1, y_j^* = 1$, $A_{ij}$ is Bernoulli with parameter $f_{|\tau_{ij}|}(A_{\tau_{ij}}) \cdot \exp(-\|z_i - z_j\|_2^2) = f_{|\tau_{ij}|}(A_{\tau_{ij}}) \cdot \exp(-\|x_i - \mu - x_j - \mu\|_2^2) = f_{|\tau_{ij}|}(A_{\tau_{ij}}) \cdot \exp(-\|x_i - x_j - 2\mu\|_2^2)$.

This completes the proof of the lemma. $\qquad\square$

Now, we provide the proof of the main theorem.

*Proof.* Since $X$ and $Y$ are independent, we have the following equalities

$$\begin{aligned}
\mathbb{P}(A_{ij}|A_{\tau_{ij}}, y_i, y_j) &= \int_{x_i, x_j} \mathbb{P}(A_{ij}, x_i, x_j | A_{\tau_{ij}}, y_i, y_j) dx_i dx_j \\
&= \int_{x_i, x_j} \mathbb{P}(x_i, x_j | A_{\tau_{ij}}, y_i, y_j) \cdot \mathbb{P}(A_{ij}|A_{\tau_{ij}}, y_i, y_j, x_i, x_j) dx_i dx_j \\
&= \int_{x_i, x_j} \mathbb{P}(x_i, x_j) \mathbb{P}(A_{ij}|A_{\tau_{ij}}, y_i, y_j, x_i, x_j) dx_i dx_j \\
&= \mathbb{E}_{x_i, x_j}[\mathbb{P}(A_{ij}|A_{\tau_{ij}}, y_i, y_j, x_i, x_j)].
\end{aligned} \tag{13}$$

The second last equality holds because in the modified model, latent vectors are independent of their labels. Using Lemma 2 in Appendix A.3 and following the analysis in (7), we have

$$
\begin{aligned}
\mathbb{E}_{x_i,x_j}[\mathbb{P}(A_{ij}=1|y_i=y_j,x_i,x_j)] &= f_{|\tau_{ij}|} \cdot (4\sigma^2+1)^{-d/2} \\
\mathbb{E}_{x_i,x_j}[\mathbb{P}(A_{ij}=1|y_i\neq y_j,x_i,x_j)] &= f_{|\tau_{ij}|} \cdot (4\sigma^2+1)^{-d/2} \cdot \exp(-\frac{4\|\mu\|_2^2}{4\sigma^2+1}).
\end{aligned}
\tag{14}
$$

Using the pairwise KL-based bound from [35, p. 428] and Lemma 1, we have

$$
\begin{aligned}
I(Y^*,A) &\leq \frac{1}{|\mathcal{Y}|^2}\sum_{Y\in\mathcal{Y}}\sum_{Y'\in\mathcal{Y}}\mathbb{KL}(P_{A|Y}\|P_{A|Y'}) \\
&\leq \max_{Y,Y'\in\mathcal{Y}}\mathbb{KL}(P_{A|Y}\|P_{A|Y'}) \\
&\leq \max_{y_i,y_j,y_i',y_j'}\binom{n}{2}\max_{i,j}\mathbb{KL}(P_{A_{ij}|A_{\tau_{ij}},y_i,y_j}\|P_{A_{ij}|A_{\tau_{ij}},y_i',y_j'}) \\
&= \max_{y_i,y_j,y_i',y_j'}\binom{n}{2}\max_{i,j}\sum_{A_{ij}}\mathbb{P}(A_{ij}|A_{\tau_{ij}},y_i=y_j)\cdot\log\frac{\mathbb{P}(A_{ij}|A_{\tau_{ij}},y_i=y_j)}{\mathbb{P}(A_{ij}|A_{\tau_{ij}},y_i'\neq y_j')} \\
&= \max_{y_i,y_j,y_i',y_j'}\binom{n}{2}\max_{i,j}\sum_{A_{ij}}\mathbb{E}_{x_i,x_j}[\mathbb{P}(A_{ij}|A_{\tau_{ij}},y_i,y_j,x_i,x_j)] \\
&\quad \cdot\log\frac{\mathbb{E}_{x_i,x_j}[\mathbb{P}(A_{ij}|A_{\tau_{ij}},y_i,y_j,x_i,x_j)]}{\mathbb{E}_{x_i,x_j}[\mathbb{P}(A_{ij}|A_{\tau_{ij}},y_i',y_j',x_i,x_j)]} \\
&= \binom{n}{2}\max_{i,j}\sum_{A_{ij}}\mathbb{E}_{x_i,x_j}[\mathbb{P}(A_{ij}|y_i=y_j,A_{\tau_{ij}},x_i,x_j)] \\
&\quad \cdot\log\frac{\mathbb{E}_{x_i,x_j}[\mathbb{P}(A_{ij}|y_i=y_j,A_{\tau_{ij}},x_i,x_j)]}{\mathbb{E}_{x_i,x_j}[\mathbb{P}(A_{ij}|y_i\neq y_j,A_{\tau_{ij}},x_i,x_j)]} \\
&< \binom{n}{2}\max_{i,j}\mathbb{E}_{x_i,x_j}[\mathbb{P}(A_{ij}=1|y_i=y_j,A_{\tau_{ij}},x_i,x_j)] \\
&\quad \cdot\log\frac{\mathbb{E}_{x_i,x_j}[\mathbb{P}(A_{ij}=1|y_i=y_j,A_{\tau_{ij}},x_i,x_j)]}{\mathbb{E}_{x_i,x_j}[\mathbb{P}(A_{ij}=1|y_i\neq y_j,A_{\tau_{ij}},x_i,x_j)]} \\
&= \binom{n}{2}\max_{i,j}f_{|\tau_{ij}|}(A_{\tau_{ij}})\cdot(4\sigma^2+1)^{-d/2}\cdot\log\left(1/\exp(-\frac{4\|\mu\|_2^2}{4\sigma^2+1})\right) \\
&= \binom{n}{2}\max_{i,j}f_{|\tau_{ij}|}(A_{\tau_{ij}})\cdot4(4\sigma^2+1)^{-1-d/2}\|\mu\|_2^2 \\
&\leq \binom{n}{2}4(4\sigma^2+1)^{-1-d/2}\|\mu\|_2^2 \\
&= 2(n^2-n)(4\sigma^2+1)^{-1-d/2}\|\mu\|_2^2.
\end{aligned}
\tag{15}
$$

By Fano's inequality [12] and by plugging (15) into (1), for the probability error to be at least $1/2$, it is sufficient for the lower bound to be greater than 1/2. Therefore

$$
\mathbb{P}(\hat{Y}\neq\bar{Y}) \geq 1 - \frac{I(Y^*,A)+\log 2}{n\log 2} \geq 1 - \frac{2(n^2-n)(4\sigma^2+1)^{-1-d/2}\|\mu\|_2^2+\log 2}{n\log 2} \geq \frac{1}{2}.
$$

By solving for $n$ in the inequality above, we obtain that if

$$
(4\sigma^2+1)^{-1-d/2}\|\mu\|_2^2 \leq \frac{n-2}{4(n^2-n)}\log 2,
\tag{16}
$$

then we have that $\mathbb{P}(\hat{Y}\neq\bar{Y})\geq\frac{1}{2}$. $\qquad\square$

## C   Directed Network Models

### C.1   Proof of Theorem 5

*Proof.* For simplicity we use the shorthand notation $o_{ji} = \sum_{k=1}^{i-1} A_{jk}$ to denote the number of directed edges from node $j$ to the first $i$ nodes. Thus we have $w_{ji} = \frac{(o_{ji}+1)(\mathbf{1}[y_i=y_j]s+1)}{\sum_{k=1}^{i-1}(o_{ki}+1)(\mathbf{1}[y_i=y_k]s+1)}$ and $0 \leq o_{ji} \leq i - j - 1$. This is because a node can never connect to its previous nodes according to the definition. Additionally $o_{ki} \leq i - m - 1$ for $k \leq m$ since the first $m$ nodes are not connected to each other.

From Algorithm 1 we can observe that $\tilde{w}_{ji} \leq \frac{1}{m}$ and $\tilde{w}_{ji} \geq \min w_{ji}$. Thus we have $\tilde{w}_{ji} \geq \min w_{ji} \geq \frac{2(\mathbf{1}[y_i=y_j]s+1)}{(i-m)(i+m-1)(s+1)}$ by assuming $o_{ji} = 0$, $o_{ki} = i - m - 1$ for every $k \leq m$, and $o_{li} = i - l - 1$ for every $l > m$.

By using the pairwise KL-based bound from [35, p. 428] and Lemma 1, we have

$$
\begin{aligned}
I(Y^*, A) &\leq \frac{1}{|\mathcal{Y}|^2} \sum_{Y \in \mathcal{Y}} \sum_{Y' \in \mathcal{Y}} \mathbb{KL}(P_{A|Y} \| P_{A|Y'}) \\
&\leq \max_{Y,Y' \in \mathcal{Y}} \mathbb{KL}(P_{A|Y} \| P_{A|Y'}) \\
&\leq \max_{Y,Y' \in \mathcal{Y}} \binom{n}{2} \max_{j,i} \mathbb{KL}(P_{A_{ji}|A_{\tau_{ji}},y_1,\ldots,y_i} \| P_{A_{ji}|A_{\tau_{ji}},y_1',\ldots,y_i'}) \\
&\leq \max_{Y,Y' \in \mathcal{Y}} \binom{n}{2} \max_{j,i} m\tilde{w}_{ji} \cdot \log \frac{m\tilde{w}_{ji}}{m\tilde{w}_{ji}'} \\
&\leq \binom{n}{2} \log \frac{1}{\frac{2m}{(n-m)(n+m-1)(s+1)}} \\
&= (n^2 - n)/2 \cdot \log \frac{(n-m)(n+m-1)(s+1)}{2m} \\
&\leq (n^2 - n)/2 \cdot \log \frac{n^2(s+1)}{8m}.
\end{aligned}
\tag{17}
$$

By Fano's inequality [12] and by plugging (11) into (1), for the probability error to be at least $1/2$, it is sufficient for the lower bound to be greater than 1/2. Therefore

$$
\mathbb{P}(\hat{Y} \neq \bar{Y}) \geq 1 - \frac{I(Y^*, A) + \log 2}{n \log 2} \geq 1 - \frac{\frac{n^2-n}{2} \cdot \log \frac{n^2(s+1)}{m} + \log 2}{n \log 2} \geq \frac{1}{2}.
$$

By solving for $n$ in the inequality above, we obtain that if

$$
\log \frac{s+1}{8m} \leq \frac{n-2}{n^2-n} \log 2 - 2 \log n,
\tag{18}
$$

or equivalently,

$$
\frac{s+1}{8m} \leq \frac{2^{(n-2)/(n^2-n)}}{n^2},
\tag{19}
$$

then we have that $\mathbb{P}(\hat{Y} \neq \bar{Y}) \geq \frac{1}{2}$. $\qquad\square$

### C.2   Proof of Theorem 6

*Proof.* From Algorithm 1 we can observe that $\tilde{w}_{ji} \leq \frac{1}{m}$ and $\tilde{w}_{ji} \geq \min w_{ji}$. Thus we have for any node $j \in \{i-m,\ldots,i-1\}$, $\tilde{w}_{ji} \geq \min_{j \in \{i-m,\ldots,i-1\}} w_{ji} \geq \frac{p(\mathbf{1}[y_i=y_j]s+1)}{m(s+1)}$ by assuming $y_i = y_k$ for every $k \in \{i-m,\ldots,i-1\}$. Similarly, for any node $j \in \{1,\ldots,i-m-1\}$, we have $\tilde{w}_{ji} \geq \min_{j \in \{1,\ldots,i-m-1\}} w_{ji} \geq \frac{(1-p)(\mathbf{1}[y_i=y_j]s+1)}{(i-m-1)(s+1)}$ by assuming $y_i = y_k$ for every $k \in \{1,\ldots,i-m-1\}$.

By using the pairwise KL-based bound from [35, p. 428] and Lemma 1, we have

$$I(Y^*, A) \le \frac{1}{|\mathcal{Y}|^2} \sum_{Y \in \mathcal{Y}} \sum_{Y' \in \mathcal{Y}} \mathbb{KL}(P_{A|Y} \| P_{A|Y'})$$

$$\le \max_{Y,Y' \in \mathcal{Y}} \mathbb{KL}(P_{A|Y} \| P_{A|Y'})$$

$$\le \max_{Y,Y' \in \mathcal{Y}} m(n-m) \max_{j \in \{i-m,\dots,i-1\}} \mathbb{KL}(P_{A_{ji}|A_{\tau_{ji}},y_1,\dots,y_i} \| P_{A_{ji}|A_{\tau_{ji}},y'_1,\dots,y'_i})$$

$$+ \binom{n-m}{2} \max_{i',j' \in \{1,\dots,i'-m-1\}} \mathbb{KL}(P_{A_{ji}|A_{\tau_{j'i'}},y_1,\dots,y'_i} \| P_{A_{j'i'}|A_{\tau_{j'i'}},y'_1,\dots,y'_{i'}})$$

$$\le m(n-m) \log \frac{1}{\frac{p}{s+1}} + \binom{n-m}{2} \log \frac{1}{\frac{m(1-p)}{(n-m-1)(s+1)}}$$

$$= m(n-m) \log \frac{s+1}{p} + \binom{n-m}{2} \log \frac{(n-m-1)(s+1)}{m(1-p)}$$

$$\le \frac{n^2}{4} \log \frac{s+1}{p} + \frac{n^2}{4} \log \frac{n(s+1)}{m(1-p)}$$

$$= \frac{n^2}{4} \left( \log \frac{(s+1)^2}{mp(1-p)} + \log n \right). \tag{20}$$

By Fano's inequality [12] and by plugging (11) into (1), for the probability error to be at least $1/2$, it is sufficient for the lower bound to be greater than 1/2. Therefore

$$\mathbb{P}(\hat{Y} \ne \bar{Y}) \ge 1 - \frac{I(Y^*, A) + \log 2}{n \log 2} \ge 1 - \frac{\frac{n^2}{4} \left( \log \frac{(s+1)^2}{mp(1-p)} + \log n \right) + \log 2}{n \log 2} \ge \frac{1}{2}.$$

By solving for $n$ in the inequality above, we obtain that if

$$\log \frac{(s+1)^2}{mp(1-p)} \le \frac{2 \log 2}{n} - \frac{4 \log 2}{n^2} - \log n, \tag{21}$$

or equivalently,

$$\frac{(s+1)^2}{mp(1-p)} \le \frac{2^{2(n-2)/n^2}}{n}, \tag{22}$$

then we have that $\mathbb{P}(\hat{Y} \ne \bar{Y}) \ge \frac{1}{2}$. $\qquad \square$