[Reviews · NeurIPS 2018]

Reviewer 1



This paper establishes recoverability results for a number of graph-based inference problems inspired by the famous stochastic block model. These inference problems are inspired by some well studied graph constructions and seem to be new, though I am not an expert on whether they have been studied before. I do not think this paper ought to be accepted, for reasons that I outline below. First, the authors should clarify whether or how they have improved on previous results on the stochastic block model. Their non-recovery bounds bounds are much weaker and it is not clear that the authors have added anything to the tight results of [21]. Second, as the authors note, the SBM is very well studied, and the whole point of that literature has been to obtain exact constants (phase transitions) for recovery. In other words, it is very easy to show non-recoverability happens at around O(1/n). As it stands, the authors have obtained these very easy results for a host of different models, but without exactly matching bounds in the other direction (or even bounds matching up to the right order, for new models), I do not think these results are especially interesting. Third, I do not think the techniques are of sufficient novelty to warrant publication. The authors emphasize their use of Fano's inequality, but this is a very standard tool. Moreover, all bounds are essentially based on the simple mutual information bound I(Y, A) \leq n^2/4 KL(p || q), which is the same bound which would arise from treating each entry in the adjacency matrix as independent. I do not think that there are any new ideas here which merit special attention.

Reviewer 2



The paper investigates information-theoretic limits for community detection with two clusters on a variety of network models using restricted ensembles and Fano's inequality. The results established in this paper yield novel bounds for the well-studied case of Stochastic Block Model, while yield the first results of their kind for the Exponential Random Graph Model, the Latent Space Model, the Directed Preferential Attachment Model, and the Directed Small-world Model, along with some dynamic models where the edge distribution not only depends on vertices but also on the edges previously generated. The paper is well-written, and the math is sound. To the best of my knowledge, this is the first paper that provides a unifying framework that yields results on a variety of network models.

Reviewer 3



This article proves information theoretic lower bounds for the community detection problem in a range of network models. For the SBM, this has been previously achieved in works of Mossel-Neeman-Sly, Coja-Oghlan, Abbe-Sandon etc. The authors of the current paper plant community structure (two randomly assigned communities) in various other models such as latent space models, preferential attachment models, small-world models etc. and prove similar information theoretic lower bounds. The proofs employ (not surprisingly) Fano's inequality in carefully restricted submodels. However, the authors only prove lower bounds, it is not clear if these are tight (it is for the SBM), and if so what (preferably polynomial time) algorithms can achieve them. There are a host of other interesting questions to ask. For example, is there an information-computation gap for larger number of communities as in the case of the SBM? I found the paper interesting and believe that it is a good contribution to the community detection literature. Here are some comments/suggestions for the authors: 1. Repetition of the sentences "we are interested in information theoretic limits. Thus, we define the Markov Chain Y* --> A --> Y. Using Fano's inequality..." before every theorem is not necessary. Say it once in the beginning perhaps. Similarly, the sentence "The use of restricted ensembles..." is repeated in page 4, line 138. 2. Page 1, line 33: "several social networks" should be "several social network models". 3. Page 3, line 98: "In a" should be "In an". 4. Page 7, line 230: "classic Small-world" should be "classic small-world phenomenon". 5. Add commas after the if clauses in the theorems (at end of the displayed equations). 6. Since your particular ERGM is essentially an SBM with different parametrization, results of Mossel-Neeman-Sly should apply. So, you should add that the corresponding lower bound is tight for this subclass. 7. Since you have more than a page left within the NIPS page limit, it would be nice if you could say something about algorithms that could be used for community extraction in these models and/or what has been done in the literature and in what parameter regimes they are successful.